# Quantification of Thermal Acclimation in Immune Functions in Ectothermic Animals

**DOI:** 10.3390/biology13030179

**Published:** 2024-03-09

**Authors:** Franziska C. Sandmeier

**Affiliations:** Biology Department, Colorado State University-Pueblo, Pueblo, CO 81001, USA; franziska.sandmeier@csupueblo.edu

**Keywords:** ecoimmunology, experimental design, immune mechanisms, thermal acclimation

## Abstract

**Simple Summary:**

This commentary focuses on experimental techniques used by researchers to understand how the immune system of an animal can acclimate, or change, to function across different body temperatures. I categorized studies as using three different, broad techniques: (1) allowing animal body temperatures to change naturally; (2) manipulating body temperatures of live animals in a controlled environment; and (3) manipulating temperatures of immune components (plasma, cells, etc.) in laboratory assays. I reviewed how the current literature used these techniques, with the conclusion that combinations of the three different techniques increased a mechanistic understanding of acclimation. In particular, cell-based techniques seem under-utilized and can lead to a greater understanding of how an animal may be changing the numbers of cells in their bodies or changing the cells themselves to function across different temperatures.

**Abstract:**

This short review focuses on current experimental designs to quantify immune acclimation in animals. Especially in the face of rapidly changing thermal regimes, thermal acclimation of immune function has the potential to impact host–pathogen relationships and the fitness of hosts. While much of the field of ecoimmunology has focused on vertebrates and insects, broad interest in how animals can acclimate to temperatures spans taxa. The literature shows a recent increase in thermal acclimation studies in the past six years. I categorized studies as focusing on (1) natural thermal variation in the environment (e.g., seasonal), (2) in vivo manipulation of animals in captive conditions, and (3) in vitro assays using biological samples taken from wild or captive animals. I detail the strengths and weaknesses of these approaches, with an emphasis on mechanisms of acclimation at different levels of organization (organismal and cellular). These two mechanisms are not mutually exclusive, and a greater combination of the three techniques listed above will increase our knowledge of the diversity of mechanisms used by animals to acclimate to changing thermal regimes. Finally, I suggest that functional assays of immune system cells (such as quantification of phagocytosis) are an accessible and non-taxa-specific way to tease apart the effects of animals upregulating quantities of immune effectors (cells) and changes in the function of immune effectors (cellular performance) due to structural changes in cells such as those of membranes and enzymes.

## 1. Introduction

Thermal changes in the environment are broadly important to the physiology and ecology of animal taxa, but particularly to ectotherms [1,2]. In recent years, ecoimmunologists have emphasized the importance of understanding thermal effects on the immune system, the microbiome, and their interaction [2,3,4]. However, clear patterns across taxa have not emerged [2,5]. Many species will experience changing thermal regimes and environmental conditions in the face of global climate change [6,7]. Ecothermic vertebrates such as amphibians and reptiles, as well as invertebrate taxa such as corals, have experienced some of the largest species declines at the intersection of climate change and disease [1,6]. Across taxa, conservation biologists have placed an emphasis on understanding how temperature, including predictable and unpredictable changes, impacts the ecoimmunology and disease ecology of host–pathogen systems [8,9,10].

In this commentary and review, I suggest that thermal studies on immune function may lack a clearly stated experimental framework and tools to understand underlying mechanisms. I hope to address three main points in methods of quantifying the thermal acclimation of immune functions, specifically in ectothermic animals. (1) I propose a framework of general mechanisms of thermal acclimation in animals and how these mechanisms can lead to a myriad of changes in immune mechanisms and the associated microbiomes. (2) I suggest experimental designs that can explicitly address these different mechanisms at a greater level of depth. Through a small review of this literature, I emphasize both strengths and possible holes in our current understanding of the thermal acclimation of immune function in ectotherms. (3) Finally, I suggest that assays based on quantifying cell-based functions, such as phagocytosis, of immune cells allow researchers to understand thermal acclimation in terms of the up- and down-regulation of cells on the level of the whole organism as well as changes in cell function.

Physiological ecologists have studied thermal acclimation in organisms for decades, including the creation of thermal performance curves for functions such as movement and metabolic rates (Figure 1a) [11]. These curves are especially useful for quantifying these functions at peak performance, the temperature at peak performance, and the thermal breadth of the function, often measured at 2/3 or 3/4 of performance (Figure 1a) [11,12,13]. Instead of using this type of function from classical physiological ecology, immune functions could include phagocytosis by immune cells, killing of bacteria in a challenge experiment, melanization and encapsulation in insects, etc. [5,14,15]. Thermal acclimation, in general, is mediated by the nervous and endocrine systems of animals on the level of the whole organism and at the cellular level [11,16,17]. At the level of the whole organism, changes in nervous or hormonal signals can effect large changes such as changes in respiration, metabolic rate, size of organs and tissues (spleen, fat bodies), and organ-specific production of cells and molecules [11,18,19]. At the cellular level, due to either organismal signals or direct reactions to changing temperature, acclimation can include changes to the cell membrane, the enzymatic structure of the cell, and changes to organelles such as mitochondria [11,19,20]. These two levels of acclimation may occur at the same time, but this also allows for differential cell-based acclimation, especially for systems as complex as the immune system [5,17,21]. For this reason, different immune mechanisms may respond with vastly different thermal performance curves and even different rates of thermal acclimation [2,5,17].

Therefore, we can visualize two general strategies that ectothermic animals can use to acclimate the function of the immune system to either increasing or decreasing temperatures (Figure 1b). They can increase or decrease immune cells or proteins, or they can change the thermal optima or thermal breadth of these cells or proteins via structural changes (Figure 1b). In reality, animals likely use a combination of these simplified strategies. Animals that have adapted to living in environments with large temperature fluctuations may also have pre-formed or constitutive components of the immune system that have a wide thermal breadth of performance [14,15,22]. In ecoimmunology, biases toward “hotter is better” were embedded into some early theories, such as the lag hypothesis, which predicts that animals need to up-regulate cells to maintain immunity at cooler temperatures [23]. However, evidence suggests that some specific immune cells do not show this pattern and have low thermal optima [14,24,25]. Overall, many species—especially animals that have evolved in temperate regions with fluctuating temperatures and varying levels of behavioral thermoregulation—may be a mosaic or composition of immune cells that range in thermal optima and breadth. Given the complexity of the immune system, including webs of interaction among cells and redundancy in important functions of immune defense [21,26], animals may have a combination of immune functions that work differentially well across their breadth of body temperatures.

The microbiome of an animal may also respond to temperature fluctuations in a similar way [27,28,29]. The turnover of species due to a change in temperature would be similar to up- or downregulating certain host cells. Similar to host cells, certain bacteria may be able to change surface and enzymatic components to survive and function in a different thermal regime [11,19]. Given the growing recognition of the importance of the microbiome in maintaining animal health, the overall thermal acclimation of the microbiome is an important consideration in understanding the acclimation of immunity [4,29,30]. The interaction of immune cells and the commensal microbial community may show further complexities as each acclimates and changes to the thermal regime of the animal [31]. Commensal microbes may also interact with pathogens in a thermally dependent manner [27]. This type of study seems relatively new, and the focus has primarily been on microbiome composition in corals and sponges [e.g., [32,33]], or in the gastrointestinal microbiome composition of vertebrates [e.g., [3,34]].

This idea that animals may function as a mosaic of cells with different thermal performances—in both host cells and their commensal microbial community—suggests that study designs also need to take this complexity into consideration. Three types of study design have commonly been used to quantify the thermal acclimation of immune responses in ectothermic animals (Figure 2). The first approach is to study the natural changes in body temperatures of animals, due to factors such as season, altitude, or latitude [24,35,36]. It is the most ecologically relevant approach but the least reductionistic and often does not provide mechanistic explanations [8]. The second approach includes bringing animals into a controlled setting to change their thermal environment [5,10,24]. Both long- and short-term acclimation can be quantified in this way, which also allows the animal to coordinate changes via the nervous and endocrine systems to what might be unpredictable changes in the environmental temperature [5]. Finally, in vitro assays of blood/interstitial fluid or isolated cells/proteins can be used in functional immune assays across an array of incubation temperatures, which can also include a step to allow for acclimation to different thermal regimes. This is the most reductionist approach, but it allows researchers to isolate the effects of temperature on a molecular or cellular level [5,35].

## 2. Literature Review

To understand the use of these approaches in the literature, I conducted a literature review using three sets of keywords in Web of Science. The purpose of this review was not to exhaustively review all the literature—some taxa-specific literature was likely missed—but to understand general patterns in the literature that emphasized contributions to the general theory of acclimation of immune responses. The search terms “immun* acclimat* temperature* ectotherm*” generated 62 hits, “Immun* acclimat* thermal* ectotherm*” generated 50, and “Immun* thermal* performance* ectotherm*” generated 44 hits, many of which were overlapping hits. Asterisks allow the search engine to find all words that include the root-term. I then sorted all references to only include papers focused on immunology in response to true acclimation to temperatures. Thus, research that quantified the temperature preference of animals after an experimental manipulation was excluded. I also excluded growing, valuable literature that only included thermal performance curves, without addressing changes to performance curves due to acclimation [e.g., [37,38]]. If papers included thermal performance curves, with acclimation at the level of the whole animal, I did include these under in vitro assays combined with the other experimental techniques. I included papers that broadly quantified changes in gene expression and protein synthesis, if these studies included specific genes known to be important to immune function. There was a clear inflection point in the number of papers published annually in and after 2017, with a consistent, increased publication rate in the most recent six years. In all, four publications were reviews, six quantified natural changes in temperature (primarily in response to season), 29 used in vivo assays that changed the temperature of the whole organism, and nine used in vitro assays. Some papers used a combination of assays and were counted multiple times.

With notable exceptions [e.g., [17]], most papers did not explicitly distinguish among these three techniques of acclimation, especially how the methodology used was directly associated with only some mechanisms of acclimation of the immune response. In general, studies of seasonal acclimation did acknowledge caveats, such as concomitant changes in hypoxia, energy limitations, and/or trade-offs with other physiological systems [e.g., reproduction: [39,40]]. Several studies combined seasonal responses with both in vivo [24] and in vitro assays [25,40]. Combining seasonal studies with these more reductionistic manipulations often reveals complex dynamics and unexpected mechanistic explanations for seasonal patterns. For example, Goessling et al. [39] combined seasonal and in vivo study designs, uncovering differential immunological responses to fluctuating temperatures across seasons. Slama et al. [25] combined seasonal and in vitro study designs and found both differential production/regulation of leukocytes and changes in phagocytic ability by cells across seasons.

Manipulating the temperature of the organism in vivo was the most common research technique. Examples from insect ecoimmunology, in particular, have been very successful in elucidating mechanisms of acclimation using this technique [17,41]. A real strength of this technique with small animals is that quantification of immune performance can also be compared to survival and clearance rates of experimental infections [17,41]. Additionally, recent research has also used an approach to quantifying the effects of fluctuating temperatures, which is a more realistic approach to quantifying the effects of a changing climate [17]. In vivo experiments can also be expanded to include thermal acclimation of the microbiome [27], but such studies still seem relatively rare.

However, in studying large vertebrates, there are several caveats to this approach. Smaller animals are expected to have faster acclimation rates due to faster heating and cooling rates and higher mass-specific metabolic rates [42]. A previous review of the physiological literature showed a biased underestimation of thermal acclimation of larger ectotherms [42]—and no animals larger than a gopher tortoise [e.g., [39]] were represented in my review of immunological acclimation in vivo. Logistically, there is the dilemma of limiting sample size to increase the number of acclimation temperatures, or vice versa [13]. Recent research has also quantified the immunological effects on captivity stress, even for captive-bred animals, which can also confound the effects of in vivo experiments [43,44]. Other researchers point to stress due to an inability to thermoregulate in the captive environment as a physiological response that is separate from the sole effects of temperature [20,45].

In vitro assays also have limitations and strengths. They do not account for an animal’s acclimation processes, mediated via nervous or endocrine control, and require functional assays. However, in combination with natural or in vivo experimental acclimation experiments, these assays are powerful and are being used to create full performance curves [25,40]. However, they seem under-used and only one study in this review first in vitro acclimated cells to different thermal regimes, before measuring performance [46]. Some strengths of these assays are that they are non-invasive and can be run across a greater range of temperatures—even including temperatures that would be lethal to the animal. By using a greater range of incubation temperatures, they can also be used to generate true performance curves that may be impossible to achieve for larger animals. In addition, in vitro assays can also be combined with challenges with a wide array of pathogens or toxins, which may be unethical or impractical in live animals. For example, combining bacteria or fungal cells with immune cells in functional assays across incubation temperatures will allow for the quantification of reactions on the scale of their actual interactions in the host. In vertebrates, such interactions commonly occur in and on the mucosa, which can be found on epidermal surfaces [bony fish, amphibians; [30]] and the respiratory, cloacal/vaginal, and gastrointestinal tracts [21,27,47].

## 3. Considerations for Research Design

A general understanding of mechanisms of immune acclimation across taxa, is largely missing from theory. I suggest that combining these experimental approaches, particularly with the inclusion of performance curves generated by in vitro assays, will help elucidate mechanisms of acclimation. For example, combining seasonal changes with an in vitro thermal performance curve quantifies ecologically relevant variation, with a mechanistic understanding of thermal optima and the breadth of performance of cellular and molecular elements of the immune response (Figure 2). Similarly, when in vitro assays are paired with experimental in vivo acclimation experiments, researchers can quantify responses on the level of the whole organism (large-scale changes in gene expression, hormone levels, etc.) and how those impact specific thermal optima and breadths of different immune functions (bacteria killing assays, antibody secretion, phagocytosis, etc.). For example, a recent review of thermal immunology in insects found different thermal optima for four measures of immune function within and among species: functional assays, immune gene expression, kinetics of enzymes, and survival of infection [17].

A final dilemma in understanding mechanisms of thermal acclimation in the immune system is differentiating between the up- and down-regulation of cells and molecules and changes in their function. Table 1 presents a list of commonly used techniques to study immune function, primarily in ecoimmunology, which has focused largely on vertebrates and insects [2,48]. While all these assays are useful in the context of understanding thermal acclimation, only functional assays that can differentiate between quantity (relative number of cells/molecules) and function (phagocytosis, microbial killing, etc.) will address whether acclimation is occurring due to the up- or down-regulation of cells/molecules or due to cellular changes such as changes to cell membranes or enzyme composition (Table 1).
biology-13-00179-t001_Table 1Table 1Common eco-immunological assays used in the literature. This table is not comprehensive, but includes most assays commonly used across vertebrate and insect taxa. Especially for in vitro assays of thermal acclimation across taxa, phagocytic assays have the advantage of being functional assays, amenable for use with both inert substances and microbes, and allowing for differentiation among the number of cells present and the relative function of those cells.Ecoimmunological AssayImmune Parameter QuantifiedFunctional AssayDifferentiation of Quantity and Function of Molecules/CellsDirect Interaction with MicrobesGeneral or Taxa-SpecificReferences (Examples)Antibody titers (ELISA)Combined amount and avidity of antibodies (can include induced and NAbs)NoNoNoVertebrate[47]Quantification of other molecules (cytokines/antioxidants/antimicrobial peptides, etc.)Levels of proteins with immune-supportNoNoNoGeneral[17,49,50]Differential immune cell countCells in interstitial fluid/bloodNo NoNoGeneral[51,52]Red blood cell agglutionation/lysisAntimicrobial proteins: complement, NAbs, lysozyme, acute phase proteins, othersYesNoNoPredominantly used in vertebrates[53]PHA-induced inflammationComponents of inflammation: leukocytes, antibodies, cytokines, chemokinesYesNoNoVertebrate[54]Fluorescence-based phagocytic assayAcidification by phagocytesYesNoYes and No (substrate dependent)General[55]Melanization/phenoloxidase activityMelanization of foreign objectsYesNoYesArthropods[2,17]Survival due to microbial challengeWhole-organism ability to stay aliveYesNoYesGeneral[2]Microbial killing assays (plasma/hemolymph-based)Antimicrobial proteins: complement, NAbs, lysozyme, acute phase proteins, othersYesNoYes General[53]Antibody secretion by cells (ELISpot)Antibody secretion by B lymphocytes YesYesNoVertebrate[56]Microbial killing assays (cell-based)Phagocytosis/entrapment by cellsYesYesYes General[53]Phagocytic assay (cell-specific or all immune-related cells)Engulfment of substrate by specific or all leukocytesYesYesYes and No (substrate dependent)General[57,58]


One of the easiest assays to use, which can be applied widely across all animal taxa, are phagocytic assays (Table 1). Since Metchnikov’s discoveries in the late 1800s, phagocytosis has been recognized as a highly conserved immune function performed by specialized cells found in the interstitial fluid and/or blood of all animal taxa [59,60]. Phagocytosis in vertebrates is also needed for adaptive immune processes, and is thus one of the central mechanisms in both innate and adaptive responses [21]. Furthermore, phagocytosis can be induced with artificial beads that are easily visualized under a light microscope or with known quantities of microbes in microbial-killing assays [58]. Simple techniques such as blood/hemolymph smears can be used to quantify cells, or hemacytometers to additionaly quantify the numbers of viable cells if the sample is not used immediately [58]. Likely because of the simplicity of using frozen plasma samples to quantify microbial killing instead of using live white blood cells, this type of assay has been under-used in ecoimmunological studies of vertebrates, with some notable exceptions [e.g., [24,35,61]].

## 4. Conclusions

The study of immune function and the associated microbiome across temperatures is fundamentally important to understanding physiological ecology in ectotherms, but also in endotherms—especially for those who experience temperature changes during hibernation or during infections that induce fever or hypothermia [21,62]. Immune function is tightly tied to homeostasis, as some of the same processes that eliminate or reduce pathogens also rid the body of damaged or old cells [21,59]. Indeed, Metchnikoff recognized this dual function of phagocytic cells [59]. For example, a recent study on Chinese giant salamanders suggested that trade-offs in cold regimes may favor organismal maintenance (including immune function and homeostasis) overgrowth, but only up to a point at which the animal enters true torpor instead of acclimating to full function at low temperatures [34]. Additionally, immune function—along with its role in homeostasis—is expected to be tied to fitness by enhancing survivorship, at least in largely R-selected species [63]. For some organisms, quantifying immune function may be an important indicator of fitness, along with acclimation responses commonly quantified by physiologists, such as muscle function, digestion rates, and aerobic metabolism [11,19,64]. While studies on birds and mammals laid a very important framework in the field of ecoimmunology, studies on immune function on ectotherms almost always need to consider temperature, unless the animal is adapted to perform at a narrow range, as might be the case for tropical or polar species [49,64,65]. I hope that the framework presented herein can help researchers design studies that hone in on different mechanisms of thermal acclimation to tease apart the effects of adaptation to thermal regimes and plasticity to both predictable and unpredictable thermal fluctuations within those systems.

## Figures and Tables

**Figure 1 biology-13-00179-f001:**
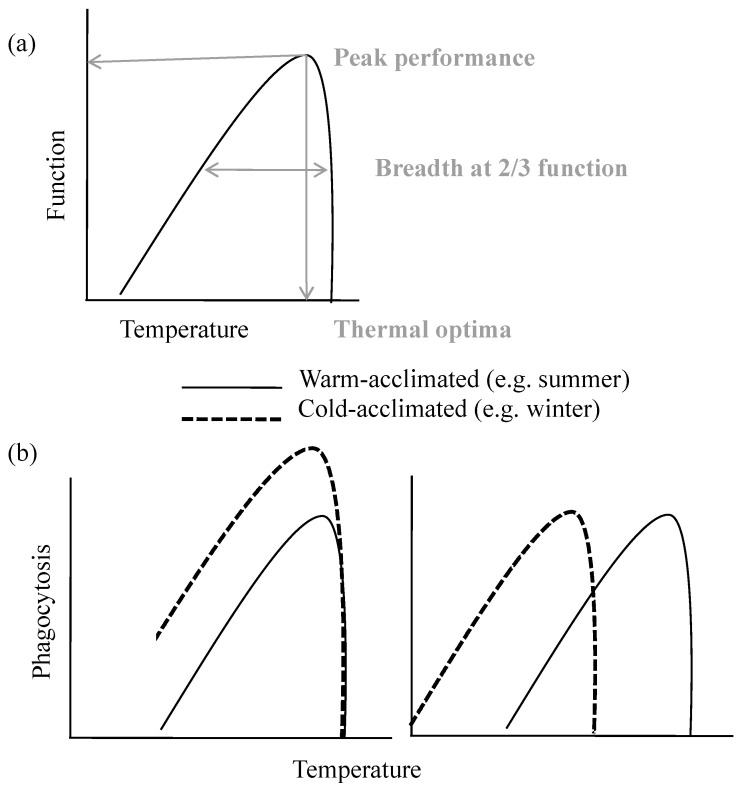
(**a**) A hypothetical performance curve, with peak performance, thermal optima, and breadth (e.g., 2/3 function) labeled. (**b**) An example of two different modes of thermal acclimation at cool temperatures in an animal, using the phagocytosis of a microbe as an example. In the left-hand panel, the numbers of phagocytic cells are increased while the thermal optimum and breadth of phagocytosis remain unchanged. In other words, function is maintained at cooler temperatures by increasing overall phagocytosis. In the right-hand panel, the numbers of cells are kept constant while the thermal optimum is shifted towards cooler temperatures.

**Figure 2 biology-13-00179-f002:**
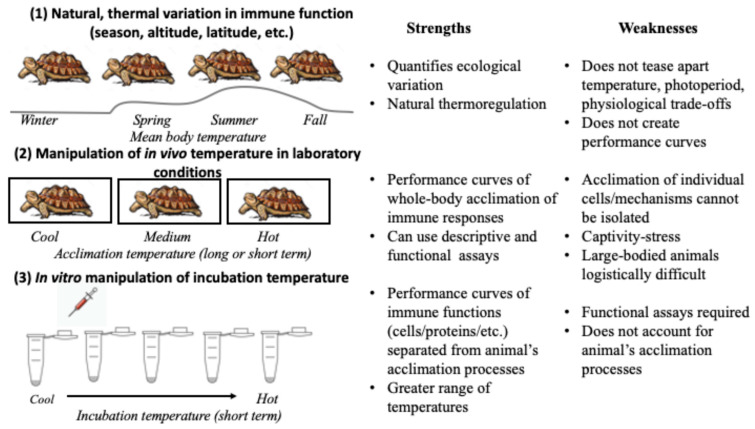
Eco-immunological study designs focused on thermal acclimation of immune responses can be grouped into three broad categories, using the example of a small vertebrate. (**1**) Natural thermal variation can be studied in wild animals due to factors such as season and geography. (**2**) Animals may be manipulated by placing them in different in vivo thermal environments in captive conditions. (**3**) Live biological samples may be incubated in vitro at different temperatures in functional immune assays. Research that combines these three categories in different ways, benefits from a combination of more holistic and reductionistic measurements and different time frames to elucidate variable mechanisms of acclimation.

## Data Availability

No new, un-reported data were generated.

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
