# Peer review of "Quantification of Thermal Acclimation in Immune Functions in Ectothermic Animals"

_biology, 2024, doi:10.3390/biology13030179_

Round 1

Reviewer 1 Report

Comments and Suggestions for Authors

Sandmeier undertook the laudable task of summarizing commonly used immunological techniques in the context of temperature acclimation.  The writing is easy to follow, and the thesis is important for the field of ecoimmunology.  However, a better synthesis of ideas may help the paper stand out more.  My comments below are designed to improve the paper, which could become a valuable contribution to the field.

General:

1. It might be nice to specifically discuss studies that have used more than one of the three types of techniques described.  These studies are valuable because they compare the qualitative results from different methods in the same experiment. 

2. Author could discuss using ecologically relevant temperature acclimation regimes (i.e., fluctuating, rather than constant temperature treatments) as these may be helpful in revealing more realistic immunological responses to climate change.

3. I understand the goal of this paper was a methodological one.  However, I was hoping to find a section summarizing our current understanding of temperature-immunity interactions using the three types of techniques described.  Do the techniques tell a similar story?  Or do they differ?  Or, perhaps, we have a general story from one technique but only parts of it from other techniques?  Related, the microbiome was mentioned several times, but there was not a summary of its role in the thermal plasticity of immune function.

4. Recommended references to include:

Adamo & Lovett (2011; J Exp Biol)—this paper examined thermal acclimation of several of the techniques reviewed in the current paper, and importantly, these authors used a fourth technique (disease resistance) considered the gold standard in ecoimmunology for invertebrates.

Butler et al. (2013; Am Nat)—this paper determined thermal performance of immune function across all major groups of vertebrates using in vitro assays.

Ferguson & Adamo (2023; J Exp Biol)—this paper reviews the insect immune system in a warming world

Specific:

L6: It might be nice to have a sentence here introducing the importance/value of immune acclimation.

L14-15: “as well as” here implies that in vitro assays are not one of “the techniques listed above”.

L24-25: Reword in active voice.

L28-29: Change “ecothermic” to “ectothermic”.

L36: It may be helpful to number these points more explicitly.  As is, there is confusion regarding “points”, “mechanisms”, and “techniques” in this paragraph.

L68: It would help to be more explicit in the rationale for the first panel of 1b.  The second (right-most) panel is intuitive (cold acclimation improves immune function at cold temperatures), but the left panel is not.

L93: Remove “also”.

L103: This sentence seems to transition from the paragraph ending on L90, rather than the preceding paragraph (ending on L102).  I recommend reordering text to better connect ideas and maintain flow.

L132: I worry that some key search terms were not used.  For example, the author describes thermal performance curves (Figs. 1 and 2), but “thermal performance” was not a search term.

L153-154: This requires context.  Insects comprise the vast majority of animal species and are generally quite small.  If the focus of this paper is on ectothermic vertebrates, then perhaps changing the title and text elsewhere is worth considering.

Author Response

Reviewer 1

Sandmeier undertook the laudable task of summarizing commonly used immunological

techniques in the context of temperature acclimation. The writing is easy to follow, and the

thesis is important for the field of ecoimmunology. However, a better synthesis of ideas

may help the paper stand out more. My comments below are designed to improve the

paper, which could become a valuable contribution to the field.

I really appreciated this review. The inclusion of added search terms and other suggestions allowed me to incorporate a broader view, to include more knowledge of invertebrate (especially insect) ecoimmunology.

General:

  1. It might be nice to specifically discuss studies that have used more than one of the three

types of techniques described. These studies are valuable because they compare the

qualitative results from different methods in the same experiment.

I have expanded this discussion in several parts of the paper - including increased citations of particularly interesting experimental designs. I also included a better description of how I tallied papers that included in vitro assays in combinations with other techniques. I realized this was a bit muddled.

  1. Author could discuss using ecologically relevant temperature acclimation regimes (i.e.,

fluctuating, rather than constant temperature treatments) as these may be helpful in

revealing more realistic immunological responses to climate change.

I added this to the section of the paper discussing in vitro assays, and added more citations from insect ecoimmunology.

  1. I understand the goal of this paper was a methodological one. However, I was hoping

to find a section summarizing our current understanding of temperature-immunity

interactions using the three types of techniques described. Do the techniques tell a similarstory? Or do they differ? Or, perhaps, we have a general story from one technique but

only parts of it from other techniques? Related, the microbiome was mentioned several

times, but there was not a summary of its role in the thermal plasticity of immune function.

These are really good points, and I added two sentences to the section “considerations for future research”. Particularly in vertebrate ecoimmunology, use of these different types of experiments does not seem to be mechanistically discussed (also in lines 177-180). I think this is a huge hole in understanding general patterns. I also added Ferguson & Adamo 2023 as an example from insect immunology that such general patterns CAN be quantified and described mechanistically. I added a sentence about the lack of inclusion of the microbiome under the discussion of in vivo assays - there is so little literature that I do not feel I can add a real summary.

  1. Recommended references to include:

Adamo & Lovett (2011; J Exp Biol)—this paper examined thermal acclimation of several of

the techniques reviewed in the current paper, and importantly, these authors used a fourth

technique (disease resistance) considered the gold standard in ecoimmunology for

invertebrates.

Butler et al. (2013; Am Nat)—this paper determined thermal performance of immune

function across all major groups of vertebrates using in vitro assays.

Ferguson & Adamo (2023; J Exp Biol)—this paper reviews the insect immune system in a

warming world

I have included these - I was not aware of Ferguson & Adamo 2023 (great paper)! I also added more in the discussion to highlight disease resistance/clearance of pathogens in vivo (luckily, I did include this in the table of techniques - but it did fall short in the rest of the paper)!

Specific:

L6: It might be nice to have a sentence here introducing the importance/value of immune

acclimation.

Added a sentence here - focusing on thermal acclimation.

L14-15: “as well as” here implies that in vitro assays are not one of “the techniques listed

above”.

Removed “as well as use of in vitro assays”, because yes, it was redundant.

L24-25: Reword in active voice.

Reworded.

L28-29: Change “ecothermic” to “ectothermic”.

Fixed.

L36: It may be helpful to number these points more explicitly. As is, there is confusion

regarding “points”, “mechanisms”, and “techniques” in this paragraph.

I numbered these main points.

L68: It would help to be more explicit in the rationale for the first panel of 1b. The second

(right-most) panel is intuitive (cold acclimation improves immune function at cold

temperatures), but the left panel is not.

Added more explanation to the figure description.

L93: Remove “also”.

Removed.

L103: This sentence seems to transition from the paragraph ending on L90, rather than the

preceding paragraph (ending on L102). I recommend reordering text to better connect

ideas and maintain flow.

I had hoped to make the point that this idea of heterogeneity in thermal performance applies to both host cells and the microbial cells (line 104).

L132: I worry that some key search terms were not used. For example, the author

describes thermal performance curves (Figs. 1 and 2), but “thermal performance” was not

a search term.

I added to the literature review by searching for “thermal* performance* immun* ectotherm*”, which yielded 44 publications. These actually included more invertebrates and corroborated the patterns detected in the literature. I updated the number of publications falling into each category - in all, this increased the literature by 12 publications.

L153-154: This requires context. Insects comprise the vast majority of animal species and

are generally quite small. If the focus of this paper is on ectothermic vertebrates, then

perhaps changing the title and text elsewhere is worth considering.

I completely agree. I changed this language to champion the large literature on arthropod thermal performance (which actually was better represented with additional search terms), and to suggest this may mostly be a limitation in studying ectothermic vertebrates. In particular, Ferguson and Adamo (2023) was a great review paper to cite - to show the advances in understanding thermal acclimation in insects.

Reviewer 2 Report

Comments and Suggestions for Authors

This review proposes a conceptual framework for adaptation including the mechanisms used by animals to thermal acclimation under climate change. Based on this analysis, the authors propose a hypothesis that is related to animal adaptation at the level of the whole organism and at the cellular level. Whereas at the organismal level, animal adaptation involves changes in nervous or hormonal signals that can cause large changes such as changes in respiration, metabolic rate and organ-specific production of cells and molecules, at the cellular level, acclimatization can be realized by phagocytic immune cells when temperature changes. In addition, the author concludes by noting that studying immune function and the associated microbiome in a temperature-dependent manner is of fundamental importance for understanding the physiological ecology of ectotherms as well as endotherms - especially those that experience temperature changes during hibernation or infections that cause fever or hypothermia. However, this scheme lacks a tissue level of adaptation, e. g., such as brown fatty tissue. It is recommended that the authors expand their analysis to include the tissue level of adaptation. In parallel to this, the addition of threshold levels of temperature change that are acceptable for ectotherm animals is also required. This last question is very important for understanding the impact of climate change on the future survival of ectotherms.

In general, the article is very useful for a wide range of readers of the journal, but it needs to be expanded and supplemented to take into account the above suggestions.

Comments on the Quality of English Language

 Minor editing of English language required

Author Response

Reviewer 2:

This review proposes a conceptual framework for adaptation including the mechanisms

used by animals to thermal acclimation under climate change. Based on this analysis, the

authors propose a hypothesis that is related to animal adaptation at the level of the whole

organism and at the cellular level. Whereas at the organismal level, animal adaptation

involves changes in nervous or hormonal signals that can cause large changes such as

changes in respiration, metabolic rate and organ-specific production of cells and

molecules, at the cellular level, acclimatization can be realized by phagocytic immune cells

when temperature changes. In addition, the author concludes by noting that studying

immune function and the associated microbiome in a temperature-dependent manner is of

fundamental importance for understanding the physiological ecology of ectotherms as well

as endotherms - especially those that experience temperature changes during hibernation

or infections that cause fever or hypothermia. However, this scheme lacks a tissue level of

adaptation, e. g., such as brown fatty tissue. It is recommended that the authors expand

their analysis to include the tissue level of adaptation. In parallel to this, the addition of

threshold levels of temperature change that are acceptable for ectotherm animals is also

required. This last question is very important for understanding the impact of climate

change on the future survival of ectotherms.

I reworded/explained how mediation on the level of the organism involves changes in tissues: increases or decreases in tissue size, due to changes in cell number (lines 96-101). Changes to cellular reactivity in those tissues can also occur. I see this as fitting into my paradigm - and indeed, I would consider leukocytes as occurring in blood “tissue”. Brown fatty tissue is only found in mammals, so that is hard to address in a paper based on ectothermic animals. I do conclude with some references to endotherms, but I think a fuller discussion distracts from the main topics of this particular commentary.

I am unsure what was meant by “threshold levels” - if it is tolerance to thermal shifts, these vary greatly by species.

I included an additional citation (Butler et al. 2013) that addresses adaptation to different thermal physiology in ectotherms (also suggested by Reviewer 1).

Reviewer 3 Report

Comments and Suggestions for Authors

Dear author, I read your review with interest. It presents an excellent overview on the approaches used to better understand thermal acclimation in ectotherms, particularly vertebrates. I did not find significant problems with the review, it reads fine (mostly, I have pointed to some terms used, which do not seem to me adequate, maybe, a matter of view, see commented ms pdf) and comes to clear conclusion. Obviously, the author favours and recommends phagocytosis assays - but why not?

So I consider this min review a good contribution to the field providing some guidance to people starting in ecoimmunology.

Author Response

Reviewer 3

Dear author, I read your review with interest. It presents an excellent overview on the

approaches used to better understand thermal acclimation in ectotherms, particularly

vertebrates. I did not find significant problems with the review, it reads fine (mostly, I have

pointed to some terms used, which do not seem to me adequate, maybe, a matter of view,

see commented ms pdf) and comes to clear conclusion. Obviously, the author favours and

recommends phagocytosis assays - but why not?

So I consider this min review a good contribution

 I changed some of the language to highlight the strengths of cell-based assays, to more fully incorporate the invertebrate literature (especially with suggestions from Reviewer 1) - but phagocytosis remains very important!

I accepted all line-by-line corrections: detailed below. I also expanded phagocytic assays to also include assays based on cellular functions (as detailed under responses to Reviewer 1).

Line 10: added “,”

Line 86: added “and thermal breadth” for clarification

Lines 108 & 121 : changed “geography” to “altitude, or latitude”

Line 140: corrected “infection” to “inflection”

Line 226: added an “s”

Round 2

Reviewer 1 Report

Comments and Suggestions for Authors

The author adequately incorporated feedback and the manuscript is improved.

Author Response

Thank you. It does not look like any further changes were suggested by the reviewer.